# Risk Assessment on the Release of *Wolbachia*-Infected *Aedes aegypti* in Yogyakarta, Indonesia

**DOI:** 10.3390/insects13100924

**Published:** 2022-10-12

**Authors:** Damayanti Buchori, Amanda Mawan, Indah Nurhayati, Aryati Aryati, Hari Kusnanto, Upik Kesumawati Hadi

**Affiliations:** 1Department of Plant Protection, Faculty of Agriculture, IPB University, Bogor 16680, Indonesia; 2Center for Transdisciplinary and Sustainability Science, Lembaga Penelitian dan Pengabdian kepada Masyarakat, IPB University, Bogor 16153, Indonesia; 3JF Blumenbach Institute of Zoology and Anthropology, Department of Animal Ecology, University of Göttingen, 37073 Göttingen, Germany; 4World Mosquito Program Yogyakarta, Centre for Tropical Medicine, Faculty of Medicine, Public Health and Nursing, University of Gadjah Mada, Yogyakarta 55281, Indonesia; 5Department of Clinical Pathology, Faculty of Medicine, Airlangga University, Surabaya 60286, Indonesia; 6Department of Family and Community Medicine, Faculty of Medicine, Public Health, and Nursing, Universitas Gadjah Mada, Yogyakarta 55281, Indonesia; 7Division of Parasitology and Medical Entomology, School of Veterinary Medicine and Biomedical Sciences, IPB University, Bogor 16680, Indonesia

**Keywords:** dengue fever, risk analysis, *Wolbachia*, Yogyakarta, eliminate dengue program

## Abstract

**Simple Summary:**

Globally, the number of dengue cases reported to the WHO increased over 8-fold over the last 2 decades, from 505,430 cases in 2000 to over 2.4 million in 2010 to 5.2 million in 2019. Reported deaths between the years 2000 and 2015 increased from 960 to 4032, affecting mostly younger age groups. The latest data in November 2021 recorded that the cumulative number of dengue cases in Indonesia was 40,759 cases (incidence rate (IR) 14.76/100,000 population) and 402 deaths (l (CFR) 0.99%). *Wolbachia*-infected *Aedes aegypti* has been hailed as a new technology that can solve dengue fever disease. Infected females are unable to transmit the dengue virus and are reproductively incompatible with uninfected males. The aim of this study is to conduct risk assessment on the release of *Wolbachia*-infected *Aedes aegypti* in Yogyakarta, Indonesia. The assessment of the risks associated with the release of *Wolbachia*-infected *Ae. aegypti* used methodology developed by the Commonwealth Scientific Industrial Research Organization (CSIRO), Australia. In this paper, the Bayesian belief network (BBN) was used as the analysis method, and combined with the discussion results and analysis data of the local expert group, the risk assessment of the release of *Wolbachia*-infected *Ae. aegypti* was carried out. The results showed that the release of *Wolbachia*-infected *Ae. aegypti* led to **negligible risk** (0.0088).

**Abstract:**

*Wolbachia*-infected *Aedes aegypti* is the latest technology that was developed to eliminate dengue fever. The Ministry of Research and Technology of the Republic of Indonesia (Kemenristekdikti) established an expert group to identify future potential risks that may occur over a period of 30 years associated with the release of *Wolbachia*-infected *Ae. aegypti*. The risk assessment consisted of identifying different hazards that may have impacts on humans and the environment. From the consensus among the experts, there were 56 hazards identified and categorized into 4 components, namely, ecological matters, efficacy in mosquito management, economic and sociocultural issues, and public health standards. There were 19 hazards in the ecological group. The overall likelihood in the ecology of the mosquito is very low (0.05), with moderate consequence (0.74), which resulted in negligible risk. For the efficacy in mosquito management group, there were 12 hazards that resulted in very low likelihood (0.11) with high consequence (0.85). The overall risk for mosquito management efficacy was very low (0.09). There were 14 hazards identified in the public health standard with very low likelihood (0.07), moderate consequence (0.50) and negligible risk (0.04). Lastly, 13 hazards were identified in the economic and sociocultural group with low likelihood (0.01) but of moderate consequence (0.5), which resulted in a very low risk (0.09). The risk severity level of the four components leading to the endpoint risk of “cause more harm” due to releasing *Wolbachia*-infected *Ae. aegypti* is negligible (0.01).

## 1. Introduction

Dengue haemorrhagic fever (DHF) is still a health problem in Indonesia, in both urban and semi-urban areas. Dengue virus (DENV) causes the wide spread of dengue fever in many regions across Indonesia. A number of cosmopolitan insects such as *Aedes aegypti*, *Ae. albopictus* and other mosquitoes [1,2,3] are the primary vectors of DENV. According to the World Health Organization (WHO) [4], DENV infections are characterized by different fever symptoms, including dengue fever, DHF accompanied by shock known as dengue shock syndrome (DSS) [5], and other unusual manifestations such as encephalopathy and cardiomyopathy. The environmental conditions and communities’ behaviors can also affect the development of DHF transmitted by *Ae. aegypti* that affects the prevalence of DHF all year long. All age groups are vulnerable to the disease. This condition is common in tropical countries, including Indonesia.

*Aedes aegypti* was first reported to be found in Indonesia in 1968 in Jakarta and Surabaya. In that year, Karyanti and Hadinegoro [6] reported the first DHF case in Jakarta and Surabaya, and the disease spread widely throughout Indonesia. The number of deaths due to DHF in 2015 was 1071, with total reported cases of 129,650. Furthermore, the incidence rate (IR) per 100,000 people in Indonesia was 50.75%, and the case fatality rate (CFR) was 0.83%; in 2016, the number of deaths was 1598, with IR of 78.85% and CFR of 0.78% [7]. Based on the recent data from the Ministry of Health Republic of Indonesia, DHF cases in Indonesia in 2020 reached 95,893, with IR 38.15 per 100,000 people and CFR 0.70%. The most cases were in West Java (18,608 cases), Bali (11,964 cases), East Java (8483), Lampung (6372), and East Nusa Tenggara (5746) [8]. Almost all the regions with high case numbers are industrial or trade centers, which have denser populations with higher mobility.

In Indonesia, the most popular government dengue vector management program is the national Breeding Site Eradication (*Pemberantasan Sarang Nyamuk*/PSN), which focuses on the “3M plus” action of covering, draining, and burying discarded water containers. Other programs include improving the water supplies, mosquito biological control using natural enemies such as mosquito-eating fish, insecticides (spraying or fogging and larval control), and also health education and community empowerment [9]. Although mosquito eradication efforts have been conducted continuously, there is still a relatively high rate of DHF cases. As a result, a new technique for controlling DHF in Indonesia by introducing *Wolbachia*-infected mosquitoes was considered [10].

*Wolbachia* are Gram-negative bacteria that cause intracellular infections in invertebrates. *Wolbachia* belong to the order Rickettsiales and are classified as strains of one species (*Wolbachia pipientis*) [11]. *Wolbachia*, particularly the strain from *Drosophila melanogaster* population (*wMel* strain), causes the ‘bendy proboscis’ phenomenon in ageing female *Ae. aegypti*. With bendy proboscis, adult females cannot penetrate into human skin to feed on blood [12]. A study conducted by Ye et al. [13] showed that *Wolbachia* can reduce the transmission potential of dengue-infected *Aedes aegypti*. Their study showed that the presence of *Wolbachia* can significantly delay the time for the mosquito saliva to become infectious, reducing the frequency of dengue virus that was expectorated by mosquitoes and lowering the virus titer in mosquito saliva. Their work also showed that *Wolbachia* can reduce the number of infectious mosquitoes in a population while also delaying the arrival of virus in mosquitoes’ saliva. 

In Indonesia, the Centre for Tropical Medicine, Faculty of Medicine, Gadjah Mada University, pioneered the use of *Wolbachia* in 2011. As a follow-up to this approach a risk assessment was conducted to evaluate the factors that influence the ecology of vectors; the social, cultural, and economic impacts of the release; the mosquito management efficacy; and the public health. The endpoint of the assessment was to address the question whether the release of *Wolbachia*-infected *Ae. aegypti* would “cause more harm” or not. Therefore, possibilities were identified concerning the likelihood that the release of *Wolbachia*-infected *Ae. aegypti* will cause more harm to the ecology of mosquitoes, dengue virus and *Wolbachia*, efficacy of mosquitoes management, standards of public health, and the social, cultural and economic conditions of the local community in release sites as well comparison of the current condition with the next 30 years.

## 2. Materials and Methods

The risk assessment core team consisted of four experts, in ecology, medical entomology, biological evolution, and medicine. In addition to the core team, 20 independent experts from universities, research institutes, nongovernment organisations, and the ministerial agencies in different areas were selected to participate in the risk assessment discussions. The expert team was composed of one virologist, two microbiologists and epidemiologists, four entomologists (medical and agriculture), one biodiversity expert, one parasitologist, one internist, one immunologist, one pediatrician, one psychologist, one public health expert, one economist, and one social scientist. The team conducted an assessment of the risks associated with the release of *Wolbachia*-infected *Ae. aegypti* using a methodology that was developed by the Commonwealth Scientific Industrial Research Organization (CSIRO), Australia [14].

Meetings and workshops were conducted to elicit opinions from experts and evidence to identify various hazards and analyze the risks associated with the release of *Wolbachia*-infected *Ae. aegypti* that may have impacts on humans and the environment. 

### 2.1. Stages in Risk Assessment

We used a risk analysis framework developed by the Australian Office to the Gene Technology Regulator (OGTR) to assess all possibilities and scenarios of unprecedented harm that may occur within the next 30 years if both female and male *Wolbachia*-infected *Ae. aegypti* were released. The assessment was conducted to evaluate the factors that influence vectors’ ecology; the social, cultural, and economic impacts of the release; the mosquito management efficacy; and the public health. The endpoint of the assessment was to address the question whether the release of *Wolbachia*-infected *Ae. aegypti* would cause more harm or not compared with the current situation within a 30-year time frame. This assessment covers several components including hazard identification, likelihood of risk, consequence of risk, and level of risk estimation (Figure 1).

A Bayesian belief network (BBN) was used for visualizing and developing the risk analysis framework and combining the expert assessment with conditional probabilities to determine the endpoint risk value. Bayes’s theorem in BBN says that future events can be predicted using any previous events that have happened [15]. BBN is a probabilistic model described in a directed acyclic graph (DAG) to demonstrate the probabilistic link between any given events [16]. It was constructed using the software package Netica© 6.09 (Norsys Software Corp. (Vancouver, BC, Canada)).

### 2.2. Problem Formulation and Hazard Identification

Experts were grouped according to the four identified components of “**cause more harm**”, namely, negative effects on ecology, decreased mosquito management efficacy, worsened public health standards, and negative sociocultural and economic impacts. Each group discussed all potential hazards leading to each component of cause more harm in the context of releasing *Wolbachia*-infected *Ae. aegypti* for the next 30 years.

The expert elicitation on hazard identification and mapping was undertaken in several steps: identification of events, determination of possible states of the events, development of the hazards list and agreed definitions, and consensus about all hazards and their definitions. Hazard and risk are often used interchangeably. Severtson and Burt [17] defined a hazard as “an act or phenomenon that has the potential to cause harm to humans or what they value” and risk as “the probability an adverse event will occur”. However, in this assessment, a hazard is a potential source of harm for humans, communities, and ecosystems. Each of the hazards (depicted as node) definitions can be found in Table 1.

### 2.3. Development of the Predictive Risk Model

A BBN was used to obtain the probabilistic relationships between events and to provide graphical representation of those events (as nodes) with possible states and a DAG from the parent node (cause) to the child node (effect). A BBN usually consists of two main components, namely, a DAG and a conditional probability table (CPT). A DAG consists of nodes and links that depicts the relationships between the variables. Here, the nodes represent the variables being observed, the hazards. Each node is connected to another node with the links (also known as arcs or edges) to show indications of conditional dependence. A link between parent node and child node showed that the nodes were functionally related or statistically correlated. Each child node (i.e., a node linked to one or more parents) contained a CPT that showed the conditional probability of the node in a specific state given by the state configurations of its parent nodes. A conditional probability is the probability of one event’s occurring if another event occurred. It was used to calculate likelihood of each node. The absence of an arc between two nodes means that no CPT can be defined. 

Bayes’s theorem was used to calculate the conditional probability at each node of an observed hazard and was applied according to the values in the CPT. The outcomes of the previous nodes were given within each node. The absolute probability as the final result was calculated by using all conditional probabilities that were previously obtained. Meanwhile, when the networks were compiled, it changed the probability distribution for the states at parent node which were also reflected in changes in the probability distribution for the states at child node.

The results of a BBN were often convincing and conclusive, even when sufficient data were not available [18]. BBN has often been used to represent knowledge and support in decision making under uncertainty [19]. It is suitable for estimating the probabilities of the occurrence of hazards caused by the release of *Wolbachia*-infected *Ae. aegypti* as a result of uncertainty (due to lack of knowledge of the long-term benefit of the presence of *Wolbachia* in natural environment).

Experts’ prior knowledge has a significant influence in hazard evaluation and the understanding of each hazard. These two factors are incorporated in the Risk Assessment using simulations that have different grades, thus ensuring that prior knowledge, assumptions and judgements are accounted in the Risk Assessment process.

### 2.4. Risk Calculation

The experts defined each hazard that may arise from the impact of *Wolbachia*-infected mosquitoes and the likelihood of each hazard based on the existing information. The consequence of a hazard was reached through discussions and consensus building based on expert assessment. Afterward, the overall risk was calculated. Here, risk was defined as an event of a particular level of severity and measured by the potential occurrence of a specific event (likelihood) multiplied by the level of resulting consequence or impact (consequence). In simple equation, risk = likelihood × consequence. 

We used the risk scale from Murray et al. [14] as the reference in determining the probability of likelihood and consequence. The scale for likelihood and consequence estimation was determined in the group discussion using a participatory process. The experts agreed on scales to score the likelihood and consequence of the identified hazards (Table 2) and the definitions for each scale (Table 3).

Discussion on the estimation of likelihoods and consequences in all groups of hazards used scales ranging from negligible to very high. Each value was calculated by considering the severity level of each hazard’s impacts on humans, the coverage and duration of the impacts, and the level of reversibility of each hazard. After determining the consequence values of each hazard, the experts then discussed the placement of each hazard into a risk matrix (Table 4).

## 3. Results

### 3.1. Hazard Identification and Mapping

The identification and mapping of the hazards as the outcome of releasing the *Wolbachia*-infected *Ae. aegypti* were based on expert elicitations and resulted in 56 hazards (nodes) excluding the end point of “cause more harm” (Figure 2). The hazards were mapped into four subcomponents of cause more harm: and altogether were combined, leading to the endpoint of “cause more harm”. The four main components were adverse impact on mosquito ecology, a lower standard of public health, decreased mosquito management efficacy, and economic and sociocultural impacts. The assessment team identified 19 ecological-related hazards including ecological effects as the endpoint (Figure 3), 12 efficacy-related hazards including mosquito management efficacy as the endpoint (Figure 4), 14 public health-related hazards including the standard of public health as the endpoint (Figure 5), and 13 economical and sociocultural hazards (Figure 6). While 56 hazards were identified (as shown in Table 1), there were two hazards (increased biting rate and transmission of non-dengue pathogens) that were shared by two groups, mosquito management efficacy and public health standard (as shown in Figure 2). Therefore, the total number of hazards became 58.

### 3.2. Likelihood

The next step of BBN in this risk assessment was discussions about the hazard likelihood of the four main components of cause more harm. The estimation yielded negligible likelihood of 1.11% (Figure 7). The likelihoods of hazards from the four components were 4.74% for ecological effects and 6.96% for the standards of public health, indicating negligible likelihood, and mosquito management efficacy and economic and sociocultural effects had likelihoods of 10.5% and 18.3%, which demonstrates a very low likelihood of risk from the hazards if *Wolbachia*-infected *Ae. aegypti* are released to suppress DENV.

### 3.3. Consequences

The expert solicitations of consequences that may arise due to the release of *Wolbachia*-infected *Ae. aegypti* were derived from a consensus on identified hazards: and among the 56 hazards, the four main components’ endpoints had moderate (ecology effects, public health, and economic and sociocultural effects) or high (mosquito management efficacy) consequences (Table 5). 

The ecology component had the most hazards, 19, including the endpoint. An amount of 18 hazards excluding the endpoint were estimated to have moderate (6 hazards), high (5 hazards), or very high (7 hazards) consequences with 57% to 90% consensus. The ecological effects as the endpoint of the ecology component had a moderate consequence with a value of 0.74. As for the mosquito management efficacy component, the expert solicitation of 10 hazards (including endpoint) resulted in a high consequence of 0.85 of the endpoints. Nine hazards, without the endpoint, were widely estimated to have very low consequence (one hazard), low consequence (two hazards), moderate consequence (two hazards), and high consequence (four hazards). 

A total of 14 hazards in the public health standard component were identified due to the release of *Wolbachia*-infected *Ae. aegypti*, leading to an endpoint of 0.5, reflecting moderate consequence (Table 5). The expert solicitation of 13 hazards without the endpoint yielded a consensus of moderate consequence for four hazards and high consequence for nine hazards. The economic and sociocultural impacts resulted in a 0.5 (moderate consequence) of this component’s endpoint. Hazards in this component were calculated to have a negligible consequence (five hazards), very low consequence (one hazard), low consequence (one hazard), moderate consequence (two hazards), and high consequence (three hazards).

### 3.4. Risk Calculation

Risk analysis workshops provided consensus concerning the estimation of the consequence and likelihood of hazards. The variables were combined to obtain the risk severity level of the four components, leading to the endpoint risk of causing more harm due to releasing *Wolbachia*-infected *Ae. aegypti*.

Overall, the expert solicitation results of the 56 hazards that may occur due to the release of *Wolbachia*-infected *Ae. aegypti* indicated an estimated high consequence (0.8) of the end point for cause more harm. The consequences for the 56 hazards ranged from negligible (5 hazards), very low (3), low (3), moderate (17), high (23), and very high consequence (6). The hazards had consensus scores of 1% to 95% for likelihood that were dominated by negligible likelihood.

Each consensus was afterwards grouped based on the risk matrix to obtain the severity levels of the risks: negligible risk, 33 hazards; very low risk, 17 hazards; low risk, 5 hazards; and moderate risk, 2 hazards (Table 5). Among the four cause more harm components, ecological influence and standard of public health were estimated to have negligible risk while efficacy of mosquito management and economic and sociocultural impacts components were estimated to have very low risk. Based on the risk estimation of 56 hazards related to the cause more harm endpoint, the release of *Wolbachia*-infected *Ae. aegypti* has negligible likelihood (0.011) and high consequence (0.8), which leads to negligible risk (0.0088).

## 4. Discussion

In this risk assessment, vector change is defined as the changes in the density, behavior, biology, and reproduction of vectors. Studies indicated that the presence of *Wolbachia* in *Ae. aegypti* mosquitoes suppresses the population size due to cytoplasmic incompatibility (CI) and suppress dengue viral transmission through the pathogen blocking effect that caused by *Wolbachia* [20,21,22]. *Ae. aegypti* infected by *wMelPop* showed a declining growth rate indicated by reduced fecundity and egg viability in *Ae. aegypti* [23,24]. In addition, *wMelPop* causes changes in the behavior of *Ae. aegypti*, as indicated by Turrey et al. [12] and Moreira et al. [25], which showed that *wMelPop*-infected mosquitoes fed on less blood meal than uninfected mosquitoes. Because the older mosquitoes spent more time in pre-probing and probing, in addition to shaking and bendy proboscis, this behavior leads to a decline in saliva production. Saliva production is associated with the DENV that accumulates in the salivary glands of *Ae. aegypti* [26,27], thus indirectly affecting DENV transmission. It also indicates that although the presence of *wMelPop* strain may cause an increased blood-feeding intensity among female adults, the mosquitoes’ ability to find blood meals also declines. Despite that, the experts assigned a relatively high score of likelihood because Weeks et al. [28] indicated that after 20 years, naturally occurring *Wolbachia*-infected *Drosophila simulans* exhibited a 10% increase in fecundity compared with that in flies that were not infected by *Wolbachia*. In other words, the bacterium characteristic changed from being parasitic to more mutualistic. 

The second low-risk hazard from the ecology component was the possibility of dengue vector replacement. In addition to *Ae. Aegypti*, mosquito species such as *Ae. Albopictus* [29], *Ae. Polynesiensis* [30], and *Ae. scutellaris* [31] are primary dengue vectors, although to date, *Ae. aegypti* are still the most effective primary vector in the transmission of DENV. History indicates that *Ae. aegypti* was first identified as the primary vector of yellow fever in 1648 in Mexico and Guadeloupe (France) [32]. The first epidemic of dengue fever transmitted by *Ae. aegypti* was recorded in 1779. Yellow fever started to become an epidemic at the beginning of the 21st century, while the dengue fever epidemic started in the 1950s. Both viruses belong to the Flaviviridae family. However, they are never found at the same time in one particular endemic area [32]. Based on this information, experts concluded that in the next 30 years, there is a likelihood that there is a very low occurrence of vector replacement because of *Wolbachia*-infected *Ae. aegypti*.

Another hazard that was concluded to have negligible risk and moderate consequence was the female-biased sex ratio. The assessment team defined this hazard as the possibility that the existence of *Wolbachia* could cause changes in *Ae. aegypti* sex ratio that might skew toward female mosquitoes, which could increase the mosquito population, which would lead to increased DHF incidence. So far, there have been no reports on the influence of *Wolbachia* on the sex ratio of *Ae. aegypti* mosquitoes or *Aedes* genus. However, Shaw et al. [33] reported that the infection of *Wolbachia* to the natural population of *Anopheles* did not influence the sex ratio of the offspring. This result outlines the relatively low influence of *Wolbachia* on the sex ratio of *Ae. aegypti* mosquitoes. To prove this, further in-depth exploration of the sex ratio of *Ae. aegypti*, after being infected with *Wolbachia*, needs to be conducted.

In the course of the hazard formulations, concerns arose regarding the possibility of the evolution of *Wolbachia* in *Ae. aegypti* that could lead to the increased fitness of filarial nematodes in mosquitoes. The consensus on increased filarial nematode fitness as a hazard indicated that there might be a negligible risk in the future. Pfarr et al. [34] concluded that *Wolbachia* that infect arthropods are distinct from *Wolbachia* that infect filarial nematodes. Pfarr et al. [34] also explained that *Wolbachia* is a parasite in arthropods but mutualists in filarial nematodes. Furthermore, arthropods and nematodes originate from different phyla, which is the risk of the evolution of *Wolbachia* present in *Ae. aegypti* in association with filarial nematodes are very low or very unlikely to happen [35].

The experts agreed that the release of *Wolbachia*-infected *Ae. aegypti* in a particular area could lead to adverse impacts on the mosquito management efficacy, but they assessed the risks as negligible (five hazards) or very low (five hazards). The five hazards with very low risk were increased difficulty to control, increased dengue virulence, household control, increased complacency, and more dengue occurrences. 

In dengue management control, sustainable vector control interventions are necessary to significantly reduce dengue transmission [4]. Community participation in dengue control needs to be continuously promoted to ensure that community members can successfully maintain their individual household environments free from dengue vectors [36]. The release of *Wolbachia*-infected *Ae. aegypti* may discourage preventive measures by the community through mosquito management. In addition, it can also increase difficulty in *Ae. aegypti* control due to the development of cryptic breeding sites [37]. 

Increasing the difficulty of controlling mosquitoes also became a critical hazard that needs to be considered mainly because it is related to *Wolbachia*-infected *Ae. aegypti* behavior changes. The changes in mosquitoes’ behavior result from the presence of *Wolbachia*, was defined by experts as changes in dengue transmission and breeding places (Table 1). However, this hazard had a very low likelihood, and moderate consequences resulted in a negligible risk. Furthermore, increased complacency at the household level due to *Wolbachia*-infected *Ae. aegypti* control may increase mosquito density, mosquito biting frequency, and a greater possibility of dengue transmission. At the community level, complacency can lead to decreased caution on the presence of *Ae. aegypti*. This particular hazard had a very low likelihood but a high consequence. This means that the hazard may have a significant influence on the success in *Ae. aegypti* mosquito management. Successful community-based vector mosquito control is influenced by numerous factors, including the community’s alert and literacy of mosquito population distribution and virus transmission rate in their respective areas [38].

Insecticide resistance is one of the hazards in the mosquito management efficacy component. At first, the experts considered this an essential issue that needed to be addressed. Since *Ae. aegypti* is a primary vector of dengue disease with a cosmopolitan range, meaning that it can be found in many tropical cities worldwide. Thus far, mosquito disease vector control has been the most effective measure in addressing dengue disease. In Indonesia, control measures have been promoted through the 3M plus (covering, draining, and burying unused water containers) program as shown in the declining DHF incidence rate [39,40]. However, available data have indicated that mosquito populations remain high [41], so that pesticides are still commonly used as an alternative measure in mosquito control in many locations in Indonesia. Therefore, the experts agreed that it has a very low likelihood with a low consequence, which resulted in a negligible risk of severity level.

Public health did not affect the release of *Wolbachia*-infected *Ae. aegypti* because the consensus from experts estimated a negligible risk with a very low likelihood and moderate consequence. The severity level of 13 hazards, excluding lower standard of public health, varied from negligible to low risk with 7 hazards have negligible risks. Increasing dengue transmission was the only hazard with a low-risk severity level. It is defined as the rate of dengue transmission increases compared with the situation before the release of *Wolbachia*-infected *Ae. aegypti*. So far, the results from studies on the rate of dengue transmission by *Wolbachia*-infected *Ae. aegypti* have indicated a decline. One of the primary factors influencing mosquitoes’ ability to transmit DENV is the extrinsic incubation period (EIP). EIP is the developmental time required for the virus to reach the mosquito’s saliva glands after an infectious blood meal. The earlier the virus appears in the saliva, the more opportunities for the mosquito to transmit DENV to humans. Ye et al. [13] reported that *wMel* lengthens the EIP, reducing the virus’s transmission frequency through the saliva. Moreover, the study also showed that *Wolbachia*-infected *Ae. aegypti* mosquito’s saliva had less DENV copy compared with wild-type mosquitoes that were not infected by *Wolbachia*. The mosquito salivary gland is the primary way for virus transmission. *Wolbachia* is mostly found in the mosquito midgut and salivary glands, both essential in transmitting the virus [42]. Therefore, the lower density of DENV in mosquito salivary glands may suggest reduced virus transmission.

The last component of this assessment was the economic, social, and cultural impact of the release of *Wolbachia*-infected *Ae. aegypti*. Initially, the group of experts focused their discussion on social and economic aspects only, but as the discussion went on, they also considered the cultural impact associated with the release of *Wolbachia*-infected *Ae. aegypti*. Since in real life, the social, behavioral, and economic factors are intertwined. The experts came to a consensus that it had a very low risk of severity level. It had 12 hazards with 7 negligible risks, 1 very low risk (scapegoating), 2 low risks (adverse media and social fear), and 2 moderate risks (class action and social conflict). Sociocultural hazards were estimated to have a higher risk than the economic ones. The sociocultural hazards may likely happen when information concerning technologies for controlling *Wolbachia* is not available in detail and does not reach all society elements, who are the main actors in community-based control. Experience from the first limited release in 2014 indicated that there were differences in opinion among the communities on whether *Wolbachia*-infected mosquitoes were safe to be released or not [39]. These differences could potentially lead to disharmony among communities. Hence, during the expert team discussion, the feedback was that awareness-raising activities are essential for preventing disharmony and conflict among the communities. There were other concerns that were raised during the discussion, e.g., the limited knowledge about the biology and evolution of *Wolbachia*, the interaction of *Wolbachia* with other species, and the nontarget impacts of the release of *Wolbachia*-infected *Ae. aegypti* on the health of the communities and the environment, which may have included the probability of an increase of filariasis as a result of the release of *Wolbachia*-infected *Ae. aegypti*. These factors need to be further understood in the future. 

The finding of *Wolbachia* was a novel breakthrough due to its innovation in addressing problematic mosquito vector control. The decline in *Ae. aegypti* mosquito populations due to cytoplasmic incompatibility (CI) and reduced vector competence is considered key in addressing the mosquito population’s problems, which have never been successfully addressed. However, the technology’s novelty needs to be assessed with caution as there is limited knowledge of the ecology of *Wolbachia*. To this point, research in some countries has indicated that *Wolbachia*-infected *Ae. aegypti* mosquitoes do not show any distinct behavior compared with the wild-type population that is not infected by *Wolbachia*. However, the future is still beyond prediction, and therefore, a risk assessment was considered necessary to ensure that all potential adverse impacts can be anticipated.

The risk assessment conducted in Indonesia estimated that over the next 30 years, there would be a negligible risk of causing more harm due to the release of *Wolbachia*-infected *Ae. aegypti*. The focus group discussion results indicated considerable critical feedback, including that continuous monitoring should be conducted after releasing *Wolbachia*-infected *Ae. aegypti* to prevent hazards identified in the assessment from happening in the natural environment. Extreme caution must be taken in responding to the result of the risk assessment. Relatively high values were assigned to the likelihoods and consequences of the identified hazards, especially the economic and sociocultural hazards (likelihood: moderate, consequence: high, risk: moderate) and social conflict (likelihood: moderate, consequence: high, risk: moderate). The experts argued that both hazards pose a danger that high value has been assigned despite the lack of scientific evidence that such hazards may occur. This indicates the high level of caution that the assessment exercised. 

## 5. Limitation

Uncertainties concerning the risks associated with the release of *Wolbachia*- infected mosquitos were thoroughly discussed. Some of the uncertainties arose because of the limitted knowledge that are available in the literatures, which then resulted in the differences in the expert judgement. Several factors can influence this different interpretation, including personal experience of the adverse impact under observation, social-cultural background and beliefs, ability to exercise control over a particular risk, access to information from different sources, and a tendency to overestimate very low risk sometimes to under-estimate very high ones. At this stage in the process, a risk must be considered a potential risk because it is unknown if it occurs in existing ecosystems. Additionally, there are probabilities of different perceptions of risks due to limited knowledge on *Wolbachia* and infected mosquitoes. The complexity of an ecosystem related to biodiversity and its interaction in the natural environment still contains many un knowns. 

## 6. Conclusions

Most of the concerns regarding the release of *Wolbachia*-infected *Ae. aegypti* stem from the lack of current knowledge on *Wolbachia*. However, scientific data have been able to address these concerns that enable experts to reach consensus on the negligible risks. The expert team conducted risk analysis based on global evidence and expert judgment resulting from comprehensive experience in health entomology, evolution ecology, public health, mosquito management, physiology, philosophy, economy, and social issues. It can be said that this assessment has covered all aspects and potential hazards of the release of *Wolbachia*-infected *Ae. aegypti* in an integrated manner. However, up-to-date knowledge should be followed and taken into consideration for the program to be able to immediately respond to changes in hazards or potential increases in risk.

## Figures and Tables

**Figure 1 insects-13-00924-f001:**
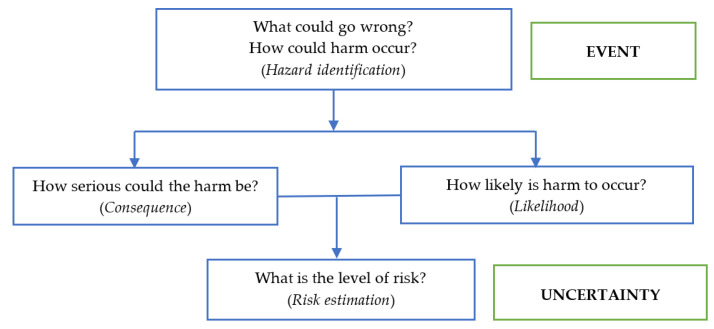
The framework of risk assessment on the release of *Wolbachia*-infected *Aedes aegypti* in Yogyakarta, Indonesia.

**Figure 2 insects-13-00924-f002:**
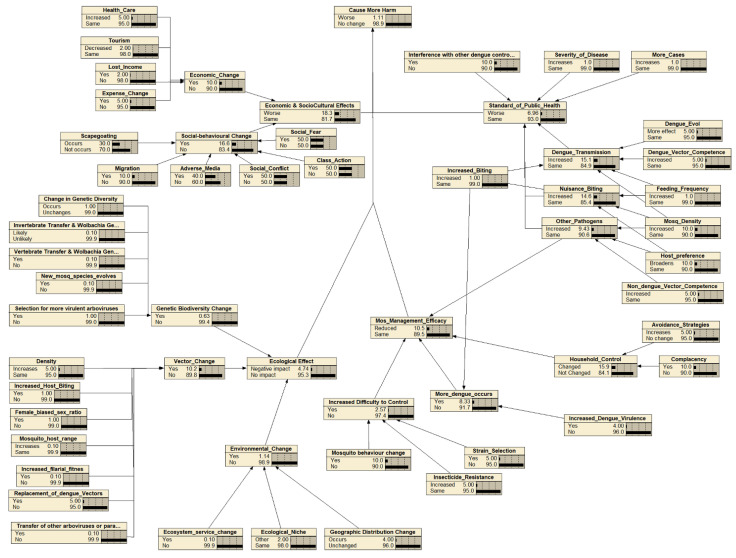
The Bayesian belief network for the endpoint “cause more harm”. Each node (box) represents probability of hazards that might occur within the next 30 years as the result of the release of *Wolbachia*-infected *Ae. aegypti*.

**Figure 3 insects-13-00924-f003:**
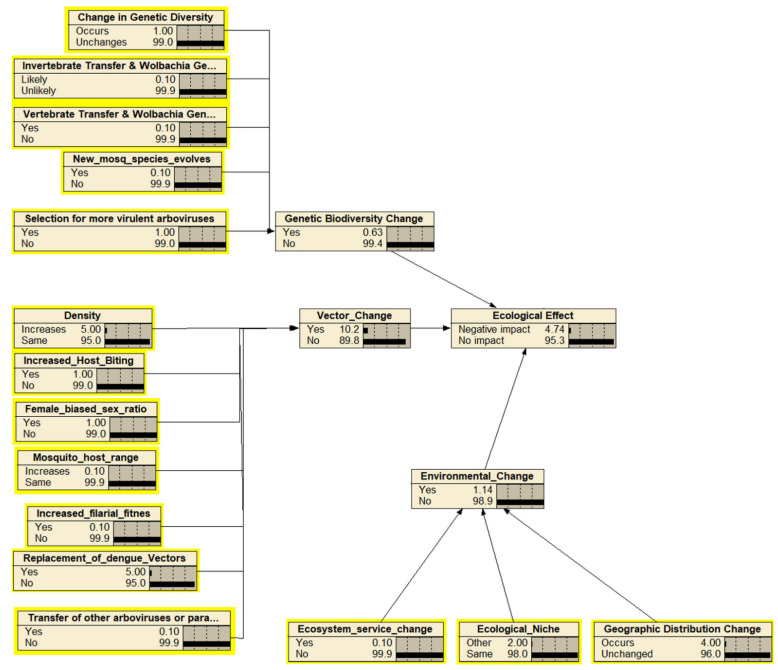
The Bayesian belief network for the endpoint “ecological effects”. Each node (box) represents probability of hazards that might occur within the next 30 years as the result of the release of *Wolbachia*-infected *Ae. aegypti*. Parent nodes are in the yellow boxes.

**Figure 4 insects-13-00924-f004:**
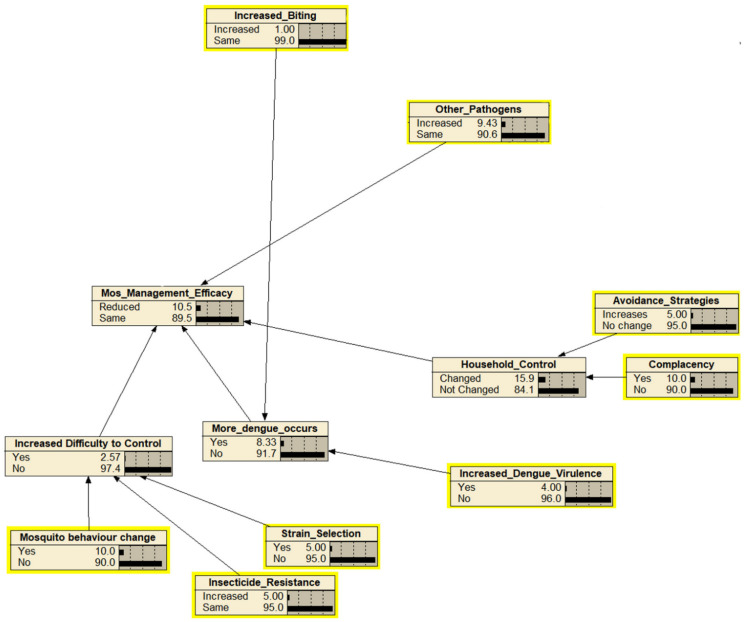
The Bayesian belief network for the endpoint “mosquito management efficacy”. Each node (box) represents the probability of hazards that might occur within the next 30 years as the result of the release of *Wolbachia*-infected *Ae. aegypti*. Parent nodes are in the yellow boxes.

**Figure 5 insects-13-00924-f005:**
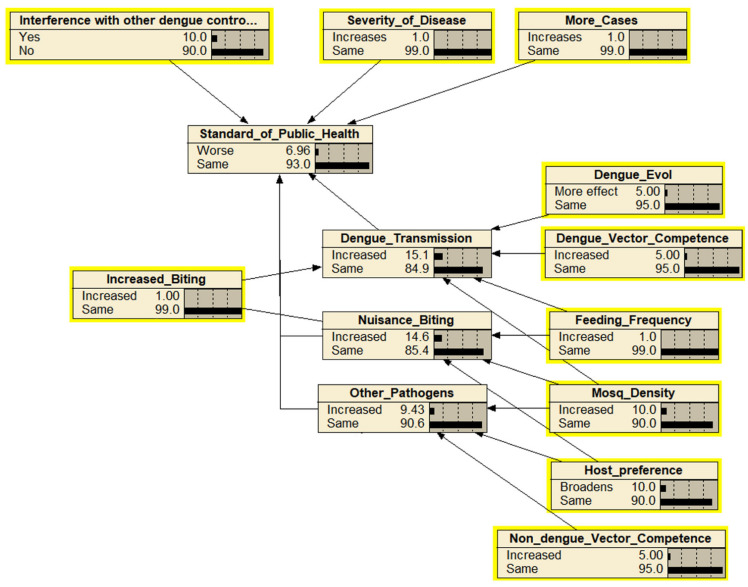
The Bayesian belief network for the endpoint “standard of public health”. Each node (box) represents the probability of hazards that might occur within the next 30 years as the result of the release of *Wolbachia*-infected *Ae. aegypti*. Parent nodes are in the yellow boxes.

**Figure 6 insects-13-00924-f006:**
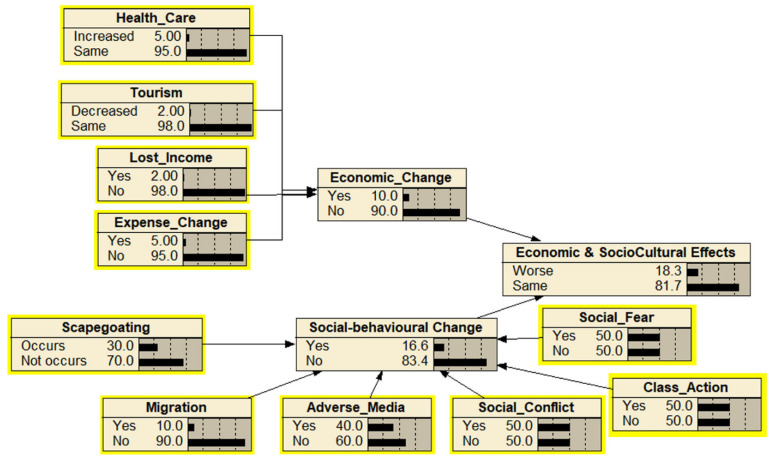
The Bayesian belief network for the endpoint “economic and socio-cultural effects”. Each node (box) represents the probability of hazards that might occur within the next 30 years as results of the release of *Wolbachia*-infected *Ae. aegypti*. Parent nodes are in the yellow boxes.

**Figure 7 insects-13-00924-f007:**
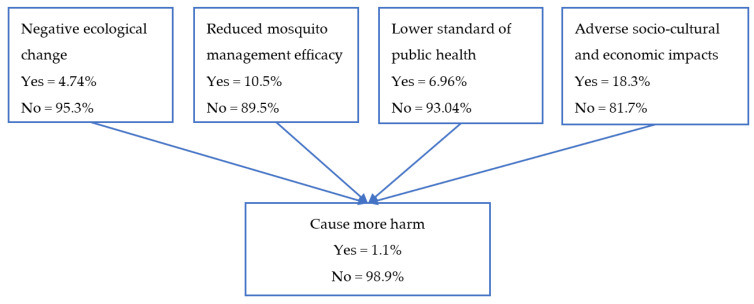
Estimated likelihood of the adverse impacts of the release of *Wolbachia* associated with four identified hazards.

**Table 1 insects-13-00924-t001:** Definition of identified hazards that may “cause more harm” upon the release of *Wolbachia*-infected *Aedes aegypti* within a 30-year time frame.

No	Hazard/Node	Definition
1	**Ecological effects**	Ecological impact of *Wolbachia*-infected *Ae. aegypti* release.
2	Genetic biodiversity change	Changes in genetic of mosquitoes, virus and *Wolbachia* in their natural habitat.
3	Change in genetic diversity	Changes in genetic diversity of *Ae. aegypti* species in nature.
4	Transfer of Wolbachia genome to invertebrates	Horizontal transfer of *Wolbachia* or some of their genomes to other invertebrates.
5	Transfer of Wolbachia genome to vertebrates	Horizontal transfer of *Wolbachia* or some of their genomes to vertebrates.
6	New mosquito species evolves	New species or strain of mosquito evolves.
7	Selection for more virulent arboviruses	Selection of more virulent arboviruses causing higher morbidity/damage and mortality.
8	Vector change	Changes in vector species, including vector density, behaviour, biology, and reproduction.
9	Increased vector density	Increased average number of mosquitoes per household due to possible changes in fecundity, longevity and vector population dynamic.
10	Increased host biting	Increased frequency of host biting by *Wolbachia*-infected *Ae. aegypti*.
11	Female biased sex ratio	Changes in sex ratio, skewed to female mosquitoes, which leads to an increase in the mosquito vector population.
12	Increased mosquito host range	Increased number of hosts other than humans enhancing the likelihood of acquiring new viruses or pathogens.
13	Increased filarial fitness	*Wolbachia*-infected *Ae. aegypti* can enhance the filarial fitness to the mosquito.
14	Replacement of dengue vectors	*Ae. aegypti* would no longer be dengue vector, replaced by other mosquito species or other organisms.
15	Transfer of other pathogens	*Ae. aegypti* may be able to transfer other arboviruses or parasites e.g., Zika or filariasis.
16	Environmental change	Changes in geographical distribution, niche of *Ae. aegypti* habitat and ecosystem services in certain areas.
17	Ecosystem service change	Changes in ecosystem structure, functions or services.
18	Ecological niche	Changes of ecological niche of *Ae. aegypti* from being a domestic species to a broader or alternative niche.
19	Geographic distribution	Changes in geographical distribution of *Ae. aegypti.*
20	**Mosquito management efficacy**	Management efficacy of *Ae. aegypti* control.
21	Increased difficulty to mosquito control	Increased difficulty in mosquito control due to changes in breeding places of *Wolbachia*-infected *Ae. aegypti.*
22	Mosquito behaviour change	Changes in behaviour of *Wolbachia*-infected *Ae. aegypti* related to dengue transmission and breeding places.
23	Increased resistance to insecticide	Increased resistance to dose and types of insecticide after *Wolbachia*-infected *Ae. aegypti* mosquitoes have been release and established.
24	Strain selection	Emergence of *Ae. aegypti* with higher vector capacity.
25	More dengue infections occur	Increased transmission of dengue virus.
26	Increased dengue virulence	Worse clinical outcomes caused by dengue infection.
27	Increased biting	Increased the probability of the biting rate of Wolbachia-infected *Ae. aegypti*.
28	Household control	Changes in dengue vector control activities by household members.
29	Avoidance strategy	Changes in normal mosquito avoidance strategies.
30	Complacency	Decreased community participation in dengue vector control due to perceived comfort and safety.
31	**Standards of public health**	The overall standard of public health.
32	Interference with other dengue control	The presence of *Wolbachia*-infected *Ae. aegypti* has caused disruption to the larva free index indicators as part of the dengue contro program.
33	Severity of disease	More severe manifestations of dengue infection, and elderly people affected by the disease.
34	More dengue cases	Increased number of dengue cases.
35	Dengue transmission	The rate of dengue transmission increases compared to the situation before the release of *Wolbachia*-infected *Ae. aegypti.*
36	Nuisance biting	Increased pest status of *Ae. aegypti*, due to increased tendency to associate with people, uninhabited houses, severity of bites and mosquito population density.
37	Other pathogens (transmission of nondengue pathogens)	Increased capability of *Ae. aegypti* to transmit pathogens other than dengue virus.
38	Dengue evolution	Dengue virus evolves so that its transmission would be more effective.
39	Dengue vector competence	*Ae. aegypti* becomes a more capable vector in transmitting dengue virus.
40	Feeding frequency	*Ae. aegypti* takes blood meal more frequently.
41	Mosquito density	Average number of *Ae. aegypti* per household would be higher.
42	Host preference	Increased variety of host animal infested with *Ae. aegypti.*
43	Nondengue vector competence	Increased vector competence as disease agents of other diseases than dengue.
44	**Economic and sociocultural impacts**	Economic and socio-cultural change due to the release of *Wolbachia*-infected *Ae. aegypti.*
45	Economic change	Decreased income and increased expenses will negatively change the economy.
46	Health care cost	The cost for health care in general will increase.
47	Tourism	Local and international tourism will be affected by the release.
48	Loss income	Individual and corporate businesses will lose lost their incomes.
49	Expense change	Increased expenses due to monitoring and controlling mosquitoes.
50	Socio-behavioural change	Negative social behaviour and deterioration of local wisdom, such as increased social isolation and decreased community participation.
51	Scapegoating	Negative collective defence mechanism as technology fails.
52	Migration	Changes in destination of migration area due to perceived safety or perceived threat.
53	Adverse media	Negative social media messages leading to concerns among the public.
54	Social conflict	Contradictory opinions in the society based on different knowledge and beliefs.
55	Class action	Legal actions from individuals, groups, communities, and community organizations.
56	Social fear	Collective mental confusions due to unintended consequences without proper assurance.

Hazards/nodes with bold letters are the four identified components of “cause more harm”.

**Table 2 insects-13-00924-t002:** Scale for likelihood and consequence estimation used for calculating the risk of identified hazards with the endpoint of causing more harm.

Scale	Negligible	Very Low	Low	Moderate	High	Very High
Probability	0–0.01	0.02–0.10	0.11–0.40	0.41–0.74	0.75–0.89	0.90–1

**Table 3 insects-13-00924-t003:** The definition of each scale that may result from each identified hazard with the endpoint of causing more harm.

Scale	Definition
Negligible	Almost no change.
Very low	Insignificant impact on human health and social economy.
Low	Very low impact or no damage to the ecosystem.
Moderate	Causes harm to human health but can be repaired, and the impact on socio-economic conditions is relatively small. The environmental damage or disturbance to local biodiversity is reversible and limited in space and time or in the amount of diversity that affected by the damage.
High	Adverse health effects are difficult to reverse but not life-threatening and have moderate socioeconomic impacts on communities.Long-term damage to the environment or disturbance to biodiversity that is still reversible.
Very high	Adverse health effects that are severe, widespread, irreversible, life-threatening, and devastating to the socioeconomic conditions.Extensive damage to the environment or disturbances to biodiversity and ecosystems, communities, or the species that survive in those ecosystems and this is not easily reversible.

**Table 4 insects-13-00924-t004:** Matrix of the risk level of each identified hazard associated with “cause more harm”.

	Consequence						
Likelihood		Negligible	Very low	Low	Moderate	High	Very high
	Negligible	Negligible risk	Negligible risk	Negligible risk	Negligible risk	Negligible risk	Very low risk
	Very low	Negligible risk	Negligible risk	Negligible risk	Negligible risk	Very low risk	Low risk
	Low	Negligible risk	Negligible risk	Negligible risk	Very low risk	Low risk	Moderate risk
	Moderate	Negligible risk	Negligible risk	Very low risk	Low risk	Moderate risk	High risk
	High	Negligible risk	Very low risk	Low risk	Moderate risk	High risk	Extreme risk
	Very high	Negligible risk	Very low risk	Low risk	Moderate risk	High risk	Extreme risk

**Table 5 insects-13-00924-t005:** Consensus of estimation of likelihood, consequence and risk (ranked by risk) for the endpoint “cause more harm”.

No	Hazard/Node	Likelihood	Likelihood Scale	Consequence Consensus	Consequence Scale	Consequence Risk	Risk Matrix Scale
1	**Ecological effect**	**0.05**	**Very low**	**0.74**	**Moderate**	**0.04**	**Negligible**
2	Genetic biodiversity change	0.01	Negligible	0.90	Very high	0.01	Very low
3	Change in genetic diversity	0.01	Negligible	0.74	Moderate	0.01	Negligible
4	Invertebrate transfer and *Wolbachia* genome	<0.01	Negligible	0.75	High	<0.01	Negligible
5	Vertebrate transfer and *Wolbachia* genome	<0.01	Negligible	0.95	Very high	<0.01	Very low
6	New mosquito species evolves	<0.01	Negligible	0.95	Very high	<0.01	Very low
7	Selection for more virulent arboviruses	0.01	Negligible	0.75	High	0.01	Negligible
8	Vector change	0.10	Very low	0.90	Very high	0.09	Low
9	Vector density	0.05	Very low	0.75	High	0.04	Very low
10	Increased host biting	0.01	Negligible	0.89	Very high	0.01	Negligible
11	Female biased sex ratio	0.01	Negligible	0.57	Moderate	0.01	Negligible
12	Mosquito host range	<0.01	Negligible	0.74	Moderate	<0.01	Negligible
13	Increase filarial fitness	<0.01	Negligible	0.75	High	<0.01	Negligible
14	Replacement of dengue vectors	0.05	Very low	0.90	Very high	0.05	Low
15	Transfer of other arboviruses or pathogens	<0.01	Negligible	0.75	High	<0.01	Negligible
16	Environmental change	0.11	Negligible	0.90	Very high	0.01	Very low
17	Ecosystem service change	<0.01	Negligible	0.74	Moderate	<0.01	Negligible
18	Ecological niche	0.02	Very low	0.74	Moderate	0.02	Negligible
19	Geographic distribution change	0.04	Very low	0.57	Moderate	0.02	Negligible
20	**Mosquito management efficacy**	**0.11**	**Very low**	**0.85**	**High**	**0.09**	**Very low**
21	Increased difficulty to control	0.03	Very low	0.90	High	0.02	Very low
22	Mosquito behaviour change	0.10	Very low	0.70	Moderate	0.07	Negligible
23	Insecticide resistance	0.05	Very low	0.20	Low	0.01	Negligible
24	Strain selection	0.05	Very low	0.20	Low	0.01	Negligible
25	More dengue infections occur	0.08	Very low	0.80	High	0.07	Very low
26	Increased dengue virulence	0.04	Very low	0.80	High	0.03	Very low
27	Household control	0.16	Low	0.60	Moderate	0.10	Very low
28	Avoidance strategies	0.05	Very low	0.10	Very low	0.005	Negligible
29	Complacency	0.10	Very low	0.75	High	0.07	Very low
30	**Standards of public health**	**0.07**	**Very low**	**0.50**	**Moderate**	**0.04**	**Negligible**
31	Interference with other dengue controls	0.10	Very low	0.50	Moderate	0.05	Negligible
32	Severity of disease	0.01	Negligible	0.80	High	0.01	Negligible
33	More dengue cases	0.01	Negligible	0.80	High	0.01	Negligible
34	Increased biting	0.01	Negligible	0.80	High	0.01	Negligible
35	Dengue transmission	0.15	Low	0.80	High	0.12	Low
36	Nuisance biting	0.15	Low	0.50	Moderate	0.07	Very low
37	Other pathogens	0.10	Very low	0.50	Moderate	0.05	Negligible
38	Dengue evolution	0.05	Very low	0.85	High	0.04	Very low
39	Dengue vector competence	0.05	Very low	0.80	High	0.04	Very low
40	Feeding frequency	0.01	Negligible	0.75	High	0.01	Negligible
41	Mosquito density	0.10	Very low	0.50	Moderate	0.05	Negligible
42	Host preference	0.10	Very low	0.85	High	0.09	Very low
43	Nondengue vector competence	0.05	Very low	0.85	High	0.04	Very low
44	**Economic and sociocultural effect**	**0.18**	**Low**	**0.5**	**Moderate**	**0.09**	**Very low**
45	Economic change	0.10	Very low	0.01	Negligible	<0.01	Negligible
46	Health care	0.05	Very low	0.01	Negligible	<0.01	Negligible
47	Tourism	0.02	Very low	0.01	Negligible	<0.01	Negligible
48	Lost income	0.02	Very low	0.01	Negligible	<0.01	Negligible
49	Expense change	0.05	Very low	0.01	Negligible	<0.01	Negligible
50	Social-behavioural change	0.17	Low	0.20	Low	0.03	Negligible
51	Scapegoating	0.30	Low	0.45	Moderate	0.14	Very low
52	Migration	0.10	Very low	0.08	Very low	<0.01	Negligible
53	Adverse media	0.40	Low	0.75	High	0.30	Low
54	Social conflict	0.50	Moderate	0.75	High	0.38	Moderate
55	Class action	0.50	Moderate	0.75	High	0.38	Moderate
56	Social fear	0.50	Moderate	0.60	Moderate	0.30	Low
57	**Cause more harm**	**0.01**	**Negligible**	**0.80**	**High**	**0.008**	**Negligible**

Hazards/nodes with bold letters are the four identified components of “cause more harm” and the endpoint “cause more harm”.

## Data Availability

The data presented in this study are available on request from the corresponding author.

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
