# Peer review of "Risk Assessment on the Release of Wolbachia-Infected Aedes aegypti in Yogyakarta, Indonesia"

_insects, 2022, doi:10.3390/insects13100924_

Round 1
Reviewer 1 Report
Dear authors,
The paper is original and have a good contribution to community science. Please to consider some corrections in attached.
Thanks

Author Response
Dear reviewers
Thank you for your review. Your review means a lot to improve the quality of our articles. Here with we sent respond to your review comment:
Point 1: Line 88-109 Please to add more information about the purpose of the work and its significance. It is necessary a connection between last two paragraphs.
Response 1: We have added the requested information, by changing the reference so that it matches the time chronology.
Point 2: Line Material and Methods: Maybe introduction? Please check.
Response 2: Reviewer 1 commented that this part is best fit in introduction. We thought this part is fit in as an introduction to this section, as the assessment team was part of the materials and methods.
As reviewer 2 suggested we added the details on the number and composition of expert teams.
Point 3: Table 1: Who is the author (s) of table 1?
Response 3: The table was a result from expert elicitation on hazard identification. The experts identified and defined each of hazards. Thus, the authors of the table is the experts or assessment team.
Point 4: Reviewer 1 comment: Please add references.
Reviwer 2 Lines 393-397, It seems that some important references are missing
Response 4: The definition of this hazard was done by the assessment team as shown in Table 1.
Reviewer 2 Report
In this paper, the Bayesian belief network (BBN) was used as the analysis method and combined with the discussion results and analysis data of the local expert group, the risk assessment of the release of Wolbachia infected Aedes aegypti was carried out. The results showed that the release of Wolbachia infected Aedes aegypti lead to negligible risk (0.0088).
Line 27, “additionally....males”, Here, the basic definition of Wolbachia cytoplasmic incompatibility (CI) is confused. Male mosquitoes carrying Wolbachia mated with female mosquitoes not carrying or carrying incompatible Wolbachia, and their offspring embryos died; When the female infects Wolbachia, its offspring can survive whether the male is infected or not.
Line 35, “negligible risk (0.008).” This is different from the result in line 315. As the most important conclusion of this paper, it is necessary to maintain the unity of the preceding and the following.
In line 57, the citations should be inserted at the end of a sentence, not at the middle or beginning of the sentence, such as lines 57, line 58, and line 66. please check the full text.
Line 57, “according to [4]”, You should accurately summarize the main content of the reference "4", rather than insert a citation to let the reader retrieve the content of the reference himself; Or, delete "according to [4]".
Line 75, “population”, This paper has a large number of word wraps, which is caused by the word control at the end of the sentence, not a mistake. But if you try to change them, it will be beneficial to the article format.
Line 102-104, One problem is the number and composition of expert teams in these different fields. I think it is necessary to clarify because it determines the rationality and authority of this assessment.
Line 118, One question worth considering is whether the risk assessment is for Wolbachia-infected females, males, or both. At present, there are generally two strategies for releasing Wolbachia infected Aedes aegypti: releasing Wolbachia infected males, or making no gender difference. Obviously, it is more acceptable to release only Wolbachia infected males, because it avoids the impact of female insects on people, such as blood-sucking and biting. In other words, the public health, ecological, economic, and pest management consequences of these two methods are obviously different. I think it is necessary for you to clarify this point at least in the materials and methods, and explain it in the discussion.
Line 149, Again, this writing is too unscientific! There are two writing types. First, you can accurately point out the author of this paper, such as “Severtson et al defined"; second, "A study defined", and then place the reference at the end of the sentence. Please check the full text, there is more than one such statement.
Line 320, What should be considered is whether the inhibition of dengue fever occurs directly through ci or due to the pathogen blocking effect of Wolbachia carried by females. CI is more to limit the number of mosquito populations, and the pathogen blocking effect is the direct reason for the dengue fever inhibitory effect.
Line 348-349, “so far...”, You need to repeatedly confirm and verify this content. A common view is that Wolbachia infection causes a variety of phenomena, such as ci, pathogen blocking, maternal inheritance, and male-killing (not present in mosquitoes). The essence of these consequences is to improve the advantage of female hosts in the population, in other words, to improve the reproductive advantage of female hosts infected with Wolbachia in the population.
Lines 380-381, It seems that some important references are missing.
Line 509, reference. In recent years, many major breakthroughs have been made in the research related to Wolbachia's infection with mosquitoes, including laboratory and field research. The types of articles include research and review. However, the references in this paper lack research result from recent years. Try to find some latest research results and cite them in the article.
Line 528-529, The exact name of the species must be in italics. Please check the full paper.
Author Response
Dear reviewers
Thank you for your review. Your review means a lot to improve the quality of our articles. Here with we sent respond to your review comment
Point 1: Line 102-104, One problem is the number and composition of expert teams in these different fields. I think it is necessary to clarify because it determines the rationality and authority of this assessment.
Response 1: We have added the details on the number and composition of expert teams.
Point 2: Line 320, What should be considered is whether the inhibition of dengue fever occurs directly through ci or due to the pathogen blocking effect of Wolbachia carried by females. CI is more to limit the number of mosquito populations, and the pathogen blocking effect is the direct reason for the dengue fever inhibitory effect.
Response 2: Thank you for your comment. Yes, the pathogen bloking indeed the direct factor that suppres dengue fever. We have change the sentence.
Point 3: Line 348-349, “so far...”, You need to repeatedly confirm and verify this content. A common view is that Wolbachia infection causes a variety of phenomena, such as ci, pathogen blocking, maternal inheritance, and male-killing (not present in mosquitoes). The essence of these consequences is to improve the advantage of female hosts in the population, in other words, to improve the reproductive advantage of female hosts infected with Wolbachia in the population.
Point 4: Reviewer 1 comment: Please add references.
Reviwer 2 Lines 380-381, It seems that some important references are missing
Response 4: The definition of this hazard was done by the assessment team as shown in Table 1.